# Statistical modeling based on structured surveys of Australian native possum excreta harboring *Mycobacterium ulcerans* predicts Buruli ulcer occurrence in humans

Koen Vandelannoote[1,2]*[†], Andrew H Buultjens[1][†], Jessica L Porter[1], Anita Velink[1], John R Wallace[3], Kim R Blasdell[4], Michael Dunn[4], Victoria Boyd[4], Janet AM Fyfe[5], Ee Laine Tay[6], Paul DR Johnson[7], Saras M Windecker[8], Nick Golding[9,10,11][‡], Timothy P Stinear[1]*[‡]

[1]Department of Microbiology and Immunology, Doherty Institute for Infection and Immunity, University of Melbourne, Melbourne, Australia; [2]Bacterial Phylogenomics Group, Institut Pasteur du Cambodge, Phnom Penh, Cambodia; [3]Department of Biology, Millersville University, Millersville, United States; [4]Health and Biosecurity, Commonwealth Scientific and Industrial Research Organisation, Geelong, Australia; [5]Victorian Infectious Diseases Reference Laboratory, Doherty Institute for Infection and Immunity, Melbourne, Australia; [6]Health Protection branch, Department of Health, Victoria, Australia; [7]North Eastern Public Health Unit (NEPHU), Austin Health, Melbourne, Australia; [8]School of Ecosystem and Forest Sciences, University of Melbourne, Melbourne, Australia; [9]Telethon Kids Institute, Perth Children's Hospital, Nedlands, Australia; [10]Curtin School of Population Health, Curtin University, Bentley, Australia; [11]Melbourne School of Population and Global Health, University of Melbourne, Melbourne, Australia

*For correspondence:
kvandelannoote@pasteur-kh.
org (KV);
tstinear@unimelb.edu.au (TPS)

[†]These authors contributed
equally to this work
[‡]These authors also contributed
equally to this work

Competing interest: The authors
declare that no competing
interests exist.

Reviewing Editor: Bavesh
D Kana, University of the
Witwatersrand, South Africa

## Abstract

**Background:** Buruli ulcer (BU) is a neglected tropical disease caused by infection of subcutaneous tissue with *Mycobacterium ulcerans*. BU is commonly reported across rural regions of Central and West Africa but has been increasing dramatically in temperate southeast Australia around the major metropolitan city of Melbourne, with most disease transmission occurring in the summer months. Previous research has shown that Australian native possums are reservoirs of *M. ulcerans* and that they shed the bacteria in their fecal material (excreta). Field surveys show that locales where possums harbor *M. ulcerans* overlap with human cases of BU, raising the possibility of using possum excreta surveys to predict the risk of disease occurrence in humans.

**Methods:** We thus established a highly structured 12 month possum excreta surveillance program across an area of 350 km² in the Mornington Peninsula area 70 km south of Melbourne, Australia. The primary objective of our study was to assess using statistical modeling if *M. ulcerans* surveillance of possum excreta provided useful information for predicting future human BU case locations.

**Results:** Over two sampling campaigns in summer and winter, we collected 2,282 possum excreta specimens of which 11% were PCR positive for *M. ulcerans*-specific DNA. Using the spatial scanning statistical tool *SaTScan*, we observed non-random, co-correlated clustering of both *M. ulcerans* positive possum excreta and human BU cases. We next trained a statistical model with the Mornington

Peninsula excreta survey data to predict the future likelihood of human BU cases occurring in the region. By observing where human BU cases subsequently occurred, we show that the excreta model performance was superior to a null model trained using the previous year's human BU case incidence data (AUC 0.66 vs 0.55). We then used data unseen by the excreta-informed model from a new survey of 661 possum excreta specimens in Geelong, a geographically separate BU endemic area to the southwest of Melbourne, to prospectively predict the location of human BU cases in that region. As for the Mornington Peninsula, the excreta-based BU prediction model outperformed the null model (AUC 0.75 vs 0.50) and pinpointed specific locations in Geelong where interventions could be deployed to interrupt disease spread.

**Conclusions:** This study highlights the *One Health* nature of BU by confirming a quantitative relationship between possum excreta shedding of *M. ulcerans* and humans developing BU. The excreta survey-informed modeling we have described will be a powerful tool for the efficient targeting of public health responses to stop BU.

**Funding:** This research was supported by the National Health and Medical Research Council of Australia and the Victorian Government Department of Health (GNT1152807 and GNT1196396).

## Editor's evaluation

This study is an important contribution to the understanding of Buruli ulcer transmission in Australia. The authors provide compelling evidence that the carriage of Mycobacterium ulcerans by possums, within their small home range, can predict cases of Buruli ulcer disease in individuals who visit those areas. While not directly relevant to the transmission of Buruli ulcer in West and Central Africa, the work will be of great interest to those studying the transmission of opportunistic environmental pathogens.

## Introduction

BU is an infection of subcutaneous tissues that can leave patients with disability and life-long deformity. The causative agent, *Mycobacterium ulcerans* is a slow-growing environmental bacterium that can infect humans after introduction through skin micro-trauma (*Portaels et al., 2009*). While the exact environmental reservoir(s) and mode(s) of transmission of BU are unresolved, they do continue to be the subject of intense research. The disease has a highly focal geographical distribution and typically occurs around low-lying marshlands and riverine areas. BU occurs mostly in tropical and subtropical areas of West and Central Africa however smaller foci are recognized in South America, Southeast Asia, and Australasia (*Röltgen and Pluschke, 2019*). In Australia, although small disease foci have been described in coastal regions of Queensland and the Northern Territory, the majority of the disease burden is found in the temperate state of Victoria, where several outbreaks have been recorded over the past three decades (*Johnson, 2019*).

In Victoria, it has been established that the median incubation period is 4.8 months with peak *M. ulcerans* transmission in humans occurring in late summer (*Loftus et al., 2018*; *Trubiano et al., 2013*). Several cases of BU infections have also been reported in native wildlife and domestic mammal species including common ringtail (*Pseudocheirus peregrinus*), common brushtail (*Trichosurus vulpecula*), mountain brushtail possums (*Trichosurus cunninghami*) (*O'Brien et al., 2014*; *Portaels et al., 2001*), koalas (*Phascolarctos cinereus*) (*Portaels et al., 2001*), long footed potoroos (*Potorous longipes*), dogs (*O'Brien et al., 2011*), cats (*Elsner et al., 2008*), horses (*van Zyl et al., 2010*), and alpacas (*Portaels et al., 2001*). These naturally acquired BU infections in animals have occurred across the same geographical regions of Victoria where BU is known to be endemic for humans.

Several studies from Victoria suggest that BU is a zoonosis that first causes epizootic disease in the local native possum populations and then spills over to humans. Focused field surveys have revealed that possums excrete *M. ulcerans* DNA in their feces (excreta) in regions known to be endemic for humans while similar surveys outside endemic areas yielded negative results (*Fyfe et al., 2010*; *Carson et al., 2014*). While *M. ulcerans* DNA could be detected at low levels in a variety of other environmental samples in these studies, by far the highest concentrations of *M. ulcerans* were found in possum excreta (*Fyfe et al., 2010*). Subsequent capture and clinical assessment of free-ranging possums validated the findings of these excreta surveys by showing that subclinical *M. ulcerans* gut

carriage in possums was common while a number of animals had laboratory-confirmed BU skin lesions and in some cases, advanced systemic disease (*O'Brien et al., 2014*). Notwithstanding the technical issues with culturing slow-growing *M. ulcerans* from complex microbial samples, a recent environmental survey identified 65% of IS*2404*-positive possum excreta were also positive by a 16S rRNA bacterial viability assay (*Blasdell et al., 2022*). Finally, whole genome sequencing and comparative genomic analysis have shown that *M. ulcerans* strains isolated from human and possum lesions are part of the same transmission cycles (*Fyfe et al., 2010*). These findings suggest that in Australia, BU is a One Health issue, with arboreal marsupial mammals representing an important environmental reservoir for *M. ulcerans*.

The present outbreak on the Mornington Peninsula in Victoria is the largest on record in Australia, with over 2,200 laboratory-confirmed cases diagnosed since 2010 (*Department of Health. Victoria A, 2022a*). Concurrent with the Mornington Peninsula outbreak, has been the emergence in 2019 of a significant cluster of BU in a suburb of the major regional city of Geelong near the Bellarine Peninsula. Regions of the Bellarine Peninsula became a BU focus beginning in the late 1990s. The unprecedented increase in human BU cases since 2010 and the rapid expansion of BU endemic areas in Victoria, including incursions to within 5 km of the Melbourne city centre (*Department of Health. Victoria A, 2022b*) have highlighted how new strategies to control transmission are urgently required. An effective BU prevention and control program requires up-to-date information on the distribution of the disease and its incidence. However, the use of traditional epidemiological surveillance methods to track the emergence and movement of BU disease foci is severely limited by the long 5 month incubation period of the disease in humans (*Loftus et al., 2018*; *Trubiano et al., 2013*), which complicates attributing disease acquisition to a particular event or location. Surveillance programs for a number of zoonotic pathogens like *Borrelia burgdorferi*, West Nile virus (*Hamer et al., 2012*), and Rabies virus (*Childs et al., 2007*) are increasingly exploring the use of wildlife sentinels to monitor disease emergence and spread. Thus here, we explore the use of systematic screening surveys of possum excreta as an early warning surveillance system to monitor BU emergence in the Mornington Peninsula and Geelong region of Victoria. We show how the detection of *M. ulcerans* DNA in possum excreta was associated with the outbreak of BU disease in humans and we use statistical modeling to explore the potential of this approach as a public health tool to predict future BU emergence.

## Methods
### Study site

The Mornington Peninsula is located 70 km south of Melbourne's city center and occupies a 750 km$^2$ area that separates Port Phillip Bay from Western Port Bay. The peninsula was one of the first areas in Victoria to be explored and settled by Europeans in the early 19[th] Century. Since then, much of the native vegetation of the peninsula was cleared under pressure from urban development although fragmented pockets of remnant wild habitat have been preserved in the Mornington Peninsula National Park and are located southwest of the peninsula (*Figure 1—figure supplement 1*). The study site overlaps with the western tip of the Mornington Peninsula and is characterized by calcareous sandy soils that support a dense coastal scrubland. As the western tip of the peninsula is primarily a local tourist hotspot known for its affluent coastal resorts, a large proportion of the houses in the region serve as temporary tourist accommodations. Many of the residential properties in the study area are spacious holiday homes, set in fenced gardens planted with shrubs and trees, which represent ideal possum habitats.

To the west of the Mornington Peninsula, and on the opposite side of the Port Phillip Bay, lies Geelong at the eastern end of Corio Bay and the left bank of the Barwon River, approximately 65 km southwest of Melbourne. Geelong has an estimated urban population of 201,924 (as of June 2018) and is the second-largest city in Victoria after Melbourne (*Australian Bureau of Statistics, 2019b*). Since 2019, BU has been considered endemic in the Geelong suburb of Belmont and surrounding areas, with local transmission suspected.

### Electronic data collection

Large-scale electronic data collection was organized using the 'Build,' 'Collect,' and 'Aggregate' tools of 'Open Data Kit (ODK),' an open-source suite of tools that were designed to build data collection

platforms (*Hartung et al., 2010*). We used ODK Build to convert paper survey forms into an ODK-compatible 'electronic form.' The ODK Collect Android app was installed from the Google Play store onto five Android budget smartphones (hereafter referred to as survey phones). ODK Collect was then configured with the custom electronic form that contained all survey questions. During sampling, surveyors worked their way through the prompts of the form and answered a wide range of question-and-answer types (*Figure 1—figure supplement 2*). The survey phones automatically sent finalized submissions over mobile data to an *ODK Aggregate* instance that was running in the cloud. Our instance of *ODK Aggregate* was hosted on the 'Google Cloud Platform' cloud provider. All data collected by the survey phones were stored and managed on this platform. *ODK Aggregate* also automatically published all new database entries to *Google Sheets*. We used a custom script (*Nenadic, 2021*) to generate dynamic maps from collected data in this Google spreadsheet using the Google Maps API. This allowed a team leader to monitor the progress of multiple teams surveying in real-time. After completing the survey all collected data was exported from the *ODK Aggregate* instance in either a tabular (csv) or a geographical (kml) format.

## Standardized roadside collection of possum excreta

Samples of possum excreta were collected along the Mornington Peninsula Road network, which is mainly made up of low-traffic, single-track paved roads and unpaved gravel tracks. Samples were collected from the ground level along the fence line of residences on grassy strips and driveways along the road. To prevent re-sampling excreta from the same possum between adjacent points a sampling spacing interval of 200 m was chosen which reflects the typical home range (radius <100 m) of these highly territorial animals (*Carson et al., 2014*). A 200 m grid pattern was laid out with the help of a custom-built battery-operated distance tracker that incorporated an Arduino microcontroller, and a GPS module, with an audio beep. When moving from sampling point A to sampling point B, the distance tracker would be reset at point A after which the device would measure 200 m as the crow flies and report decreasing distance to point B by increasing the beeper's intermittent beeping rate. ODK Collect was used to capture the location of the new sampling point using the survey phone's GPS (accuracy <8 m) after which the surveyor's name was recorded. Surveyors then used the app to log a time point when they started looking for possum excreta. The search was restricted to a 50 m radius around the sampling point and was terminated in case no excreta could be found after a pre-allotted time of 5 min. Surveyors logged another time point when they discovered the first fecal pellet. Fecal material (excreta) from each sampling location was stored in separate sterile re-sealable plastic zipper bags that had been pre-labeled with a barcode. If possible, up to three excreta samples were sampled and, in case more were available, the freshest most intact excreta were selected. ODK Collect was then used to take a picture of the sampling site and scan the barcode on the zipper bag, both using the survey phone's camera. Following this, the collected excreta were used to distinguish and record the presumed species of possum based on distinctive morphological characteristics of the excreta from each species (*Figure 1—figure supplement 2*). Subsequently, completed forms were marked as finalized and uploaded to the cloud. In case no excreta material was found on a particular sampling location after 5 min, ODK Collect forked to the end of the survey where the form likewise was marked as finalized and submitted. A video was produced and uploaded to YouTube to illustrate the steps described above (*ODK, 2022*). Samples were transported at 4 °C to the laboratory where they were stored at –20 °C prior to further processing.

Several sampling missions were organized from December 19, 2018 to March 14, 2019 and are hereafter referred to as the 'summer survey.' Later, between May 28 and September 19, 2019, we attempted to revisit most of the locations we sampled during summer, a period that is hereafter referred to as the 'winter survey.' To facilitate the sampling effort during winter, the above-mentioned electronic distance trackers were reprogrammed with a predetermined grid of sampling points from the summer survey. Consequently, the trackers now helped surveyors relocate to the nearest sampling point using the same auditory cue system. During the winter survey, we also tried to determine the usefulness of surveying at a higher resolution by sampling along a 50 m grid pattern in three small regions. Apart from this, standardized ODK Collect-based roadside collection was performed identically. Another survey mission centered on the Geelong area was conducted in the early half of 2020 (January 16 through to the April 28, 2020) using the above mentioned surveying methodology with a 200 m grid.

## DNA extraction, *M. ulcerans* IS*2404* real-time PCR

For samples collected from the Mornington Peninsula, microbial genomic DNA (gDNA) was extracted from possum excreta using the DNeasy PowerSoil HTP 96 Kit (Qiagen Cat# 12955–4) following the manufacturer's protocols just prior to the addition of solution C4, whereupon DNA was subsequently purified from 200 µl of the lysate using the QIAsymphony DSP Virus/Pathogen extraction kit (Qiagen Cat# 937036) on the QIAsymphony automated platform. Extraction included two rounds of mechanical homogenization for 45 s at 1800 rpm on a FastPrep-96 instrument (MP Biomedicals). DNA extraction negative controls (distilled water) were included at a frequency of 5–10% per batch of samples and processed blinded (*Figure 1—figure supplement 3*). The extraction method for the possum excreta survey of the Geelong region followed a similar procedure but with some modifications as described (*Blasdell et al., 2022*).

Real-time PCR assays targeting IS*2404* multiplexed with an internal positive control (IPC) (Life Technologies Cat# 4308323) were carried out as described before (*Fyfe et al., 2007*). Briefly, IS*2404* real-time PCR mixtures contained 10.0 µl of 2 x SensiFast Probe NO-ROX mix (BioLine Cat# BIO-86005), 3.2 µl of nuclease-free water, 0.8 µl each of 10 µM IS*2404* TF and IS*2404* TR primers, 0.8 µl of 5 µM IS*2404* TP probe, 2.0 µl TaqMan Exogenous IPC MIX, 0.4 µl TaqMan Exogenous IPC DNA, and 2.0 µl of DNA extract in a final reaction volume of 20 µl. No-template controls (NTC) and positive *M. ulcerans* gDNA controls were also included in every real-time PCR run (*Figure 1—figure supplement 3*). Amplification and detection were performed with the Light Cycler 480 II (Roche) using the following program: 1 cycle of 95 °C for 5 min, and 45 cycles of 95 °C for 10 s, and 60 °C for 20 s.

Inclusion of adequate extraction and real-time PCR controls, as well as IPCs, was essential for ensuring the accuracy and reliability of our environmental molecular surveys. Negative extraction and negative real-time PCR controls were included to ensure there were no false positives or contaminations present in respectively the extraction runs and the real-time PCR IS*2404* assays. Included positive real-time PCR controls, on the other hand, contained a known quantity of the IS*2404* target and were included to verify that the real-time PCR assay could detect the IS*2404* target when present in the sample. Finally, IPCs – also known as 'spud' assays – were included to monitor for inhibition during real-time PCR reactions, which (if not properly monitored) can result in false negatives or poor reproducibility of results.

## Geographical data acquisition and spatial analysis

The 2011 Victorian mesh block boundaries dataset and the 2011 Victorian mesh block census population counts dataset was downloaded from the Australian Bureau of Statistics (ABS) website (*Australian Bureau of Statistics, 2019a*). Mesh blocks are relatively homogeneous statistical units and represent the smallest geographical unit for which publicly available census data are tabulated by the ABS. The mesh block digital boundaries dataset is based on Australia's national coordinate reference system, the Geocentric Datum of Australia (GDA94). Spatial information was analyzed and edited in the geographic information system (GIS) software QGIS v.3.16.7 (*QGIS Development Team, 2021*). The 2011 census population count spreadsheet was joined to the mesh block boundaries using the unique mesh block IDs. The centroid of all polygon mesh blocks was determined and their latitude and longitude in GDA94 were calculated. The state-wide geometric dataset was then down sampled to a more manageable size (3,238 mesh blocks) for subsequent spatial statistical analysis by reducing it to the ABS level 2 Statistical Areas (SA2) that encompasses the Mornington Peninsula (Point Nepean and Rosebud – McCrae). A second geographical dataset was prepared for the Geelong area by selecting the SA2 areas for that region (Belmont, Corio - Norlane, Geelong, Geelong West - Hamlyn Heights, Grovedale, Highton, Newcomb - Moolap, Newtown, and North Geelong - Bell Park). All GPS positions of sampling points visited during the excreta surveys were added to this GIS and projected to GDA94. QGIS v.3.16.7 (*QGIS Development Team, 2021*) was used to generate the figures of the excreta survey results and the geographical distribution of BU cases in the Mornington Peninsula.

## Human BU cases

The Victorian Department of Health (DH) made a de-identified database extract available of all BU patients that were notified in the state between January 1, 2019 and December 31, 2020. A human case of BU is defined here as a patient who presented with a clinical lesion suggestive of BU in which *M. ulcerans* DNA was detected in a laboratory using IS*2404* real-time PCR (*Fyfe et al., 2007*; *Ross*

*et al., 1997*). Note that BU has been a 'notifiable condition' in Victoria since 2004 and as of January 1, 2011 DH has been collecting enhanced BU surveillance data in a centralized notifiable disease database using custom collection forms (*Victorian Government, 2022*). DH extracted the data used in this study from this database and then geocoded and de-identified it at an aggregated mesh block level. As mesh blocks typically encompass between 30 and 60 dwellings the extract of the notifiable disease database was effectively anonymized. Variables used in the analysis included: the date of notification, date of first symptoms onset, and mesh block of address of residence at the time of notification.

We selected two populations of notified BU patients suspected of having been infected with *M. ulcerans* in the Mornington Peninsula during an 'exposure interval' that aligned with the organized possum excreta surveys. The reported onset of disease was used to infer this exposure interval based on the mean incubation period of BU in Victoria of 143 days (Inter-Quartile Range (IQR) 101–171) (*Loftus et al., 2018*). We define the incubation period here as the time between exposure to an endemic region and symptom onset. Patients were suspected of having been infected with *M. ulcerans* in the Mornington Peninsula if they were either a resident of the peninsula or if they visited the area and had not reported recent (<12 months) contact with any other known BU endemic areas in the state (*Figure 2—figure supplement 1*). A population of cases whose exposure interval overlapped with the survey of the Geelong region was also selected (*Figure 5—figure supplement 1*).

## Statistical analysis

### Basic statistical testing

Statistical analyses were performed using R v4.0.3 (http://www.R-project.org/). Comparison of excreta IS*2404* positivity between sampling season or possum species was done using Fisher's exact test. Comparison of mean Ct (IS*2404*) values between sampling season or possum species was done with an unpaired *t*-test while assuming equal variance (checked with an F-test).

### Spatial scan statistics

We used SaTScan v9.7 (*Kulldorff, 1997*) to analyze surveillance data with discrete spatial scan statistics. SaTScan tackles this by progressively 'scanning' a circular window of variable size across space while noting the number of observed and expected observations inside this window at each location. For each scanned circular window, a log-likelihood ratio (LLR) statistic is calculated based on the number of observed and expected cases within and outside the circle and compared with the likelihood under the null hypothesis. We investigated Mornington Peninsula surveillance data both of (i) notified human BU disease and of (ii) epizootic spread of *M. ulcerans* in possum excreta. For each dataset, we accepted both primary and secondary clusters, if (i) their corresponding p-values were less than 0.005 and (ii) secondary clusters did not overlap geographically with previously reported clusters with a higher likelihood. P-values were based on 9999 Monte Carlo simulations for each dataset, as suggested by *Kulldorff, 1997*. SaTScan parameter and input data files were made available on a GitHub Repository (*Buultjens, 2023*).

We performed geographical surveillance of human BU disease by detecting spatial disease clusters and assessing their statistical significance. To achieve this, we applied the Poisson probability model to our mesh block-level data of notified BU case counts, which arose from a background population at risk that was derived from the 2011 population census. We limited the maximum cluster size to 10% of the total population at risk (corresponding to 14,481 inhabitants) so that defined areas of risk would remain within a manageable size for targeted BU prevention campaigns with limited resources. Under the null hypothesis, BU cases are homogeneously distributed over the Peninsula. Under the alternative hypothesis, there are geographical areas with higher rates of BU than would be expected if the risk of contracting BU was evenly distributed across the Peninsula.

We used the Bernoulli probability model to scan for spatial clusters with high rates of *IS2404* positivity in sampled possum excreta. The Bernoulli model is proper here as excreta *IS2404* positivity is a variable with two categories. The maximum cluster size was set to 50% of the population size. Under the null hypothesis, excreta *IS2404* positivity is homogeneously distributed over the surveyed region. Under the alternative hypothesis, there are clusters where the *IS2404* positivity rate is higher than in regions outside of these clusters.

## Predictive modeling

To prospectively predict the occurrence of human BU cases, a statistical model was calibrated using excreta positivity and cases whose exposure period overlapped with the excreta survey periods. Specifically, cases were included if the exposure interval overlapped with the date that excreta were collected. As was stated for the abovementioned analyses, an exposure interval based upon the IQR which covered the central 50% of observations (101–171 days) was used. This approach is expected to provide a reliable estimate of the typical incubation period. To assess the full spectrum of incubation periods observed in Victoria, including outliers and extreme values, we also conducted the same analysis using the range of observed incubation periods (61–277 days). This allowed us to examine the potential impact of atypical cases on our results. Here the model was fitted separately for the summer and winter seasons with the objective to predict if a given mesh block will contain one or more human BU cases. The metric used to evaluate predictive performance was the area-under-the-curve (AUC) which is the degree of separability that describes how capable a model is at classifying mesh blocks where cases occur and those where cases don't occur. AUC values range from 0 to 1, with an AUC of 1 indicating a perfect record of classification while a value of 0.5 depicts a model that has no classification capacity. All scripts used in the statistical analytical pipeline have been made available in a GitHub Repository (**Buultjens, 2023**) and made use of the following R packages: *flexclust* (**Leisch, 2006**), *raster* (**Hijmans et al., 2022**), *readxl* (**Wickham and Jennifer, 2022**), sf (**Pebesma, 2018**), and *tidyverse* (**Wickham et al., 2019**).

We built a custom statistical model to predict the probability of observing one or more human BU cases as a function of the distance-weighted prevalence of MU positivity in nearby excreta. That is, the probability of observing a BU case in each location is greater if nearby excreta are MU-positive. One approach to modeling the risk to humans would be to construct a geostatistical model of the prevalence of MU in excreta, i.e., a spatially continuous estimate of excreta positivity for all areas using e.g., model-based geostatistics (**Diggle and Ribeiro, 2007**). However, to use such a complex model for the prediction of human cases to guide interventions would require a re-estimation of the model and its many location-specific random-effect parameters after every trapping round. This would be burdensome and limit the application of the model in a public health setting. Instead, we construct a simpler model with two parameters that calculates the local prevalence around a focal location (e.g. the centroid of a small district) as the weighted mean of the excreta positivity in all samples, with those weights decaying with increasing distance from the focal location. The statistical model we use is as follows:

$$
\begin{aligned}
y_i &\sim Bernoulli\left(p_i\right) \\
p_i &= 1 - e^{-N_i} \\
N_i &= I_i P_i \\
I_i &= \beta \sum_{j=1}^{J} w_{ij} x_j \\
w_{ij} &= \frac{w_{ij}^*}{\sum_{j=1}^{J} w_{ij}^*} \\
w_{ij}^* &= e^{-\frac{1}{2}\left(d_{ij}/\sigma\right)^2}
\end{aligned}
\tag{1}
$$

where $y_i$ is an indicator variable for whether a human BU case was detected at location $i$ in the subsequent period, modeled as a single binomial sample with probability $p_i$, calculated as the probability of observing one or more cases if case numbers in that location were Poisson-distributed with an expectation of $N_i$ cases; the product of the expected incidence $I_i$ and the population $P_i$ at each location. The incidence is modeled as the product of a positive-constrained scale parameter $\beta$, and the distance weighted average of the MU positivity $x_j$ (coded as a 1 for MU-positive or a 0 for MU-negative) across all excreta sampling locations $j$ in $J$. The applied weight is a normalized Gaussian function of the Euclidean distance $d_{ij}$ between human case location $i$ and excreta collection location $j$, with the range of the Gaussian function given by the positive-constrained parameter $\sigma$. The model is, therefore, parameterized by only two parameters: $\beta$ which controls the absolute probability of observing a BU case in any given location, and $\sigma$ which controls the distance at which excreta MU positivity is predictive of BU cases – if $\sigma$ is small, only nearby excreta MU positivity is predictive of BU cases, whereas if $\sigma$ is large, MU positivity in excreta collections further away are also informative. These two parameters are estimated by maximum likelihood using the Nelder-Mead optimization routine in R,

with five random starts. These two estimated parameters can then be used to predict the probability of BU case occurrence $p_i$ in new places and sampling rounds, by combining them with a new excreta sampling dataset.

Note that this model could easily be modified to be fitted to, and therefore predict the number $N_i$ of BU cases at a given location. However, given the comparatively low number of reported cases in each affected mesh block in our study region, the probability of the presence of one or more cases is likely to be a more useful metric for prioritization of interventions. In addition, the estimation of the conditional variance parameter of such a count model would add considerable complexity to the model fitting process and limit its utility for applied public health prioritization.

### Cross-validation model development

We first validated the predictive ability of the statistical model using a cross-validation approach using the excreta survey data from the Mornington Peninsula region. Here the mesh block dataset was split into three spatially contiguous regions so that spatial block cross-validation could be performed (*Figure 4—figure supplement 1*). The model was fitted on each possible 2/3 of the data and then used to predict the probability of observing one or more BU cases in the remaining 1/3.

### Predictions on unseen data

The model was then fitted to the entire Mornington Peninsula dataset and used to make predictions to a previously unseen dataset of the Geelong excreta survey and human BU case data. The Geelong survey was conducted in early 2020 (January 16 through to the April 28, 2020), with there being just three mesh blocks that had cases with exposure intervals overlapping with the Geelong survey period. Geelong is a city of 200,000 inhabitants, 65 km southwest of Melbourne (*Figure 1—figure supplement 1*).

### Alternative model

Null models were established for the Mornington Peninsula and Geelong datasets that used the incidence of human BU cases in mesh blocks in the year preceding to predict the likelihood of a mesh block containing a case. The null models were included as alternative models that were not based on any insights from the excreta surveys, and represent the type of inference that could reasonably be made to anticipate future case occurrence from epidemiological data alone. From a statistical perspective, the previous year's incidence null models can be considered as a different option for out-of-sample intercept-only models for model performance comparison purposes.

### Ranking mesh blocks to inform *M. ulcerans* transmission risk assessments

To provide a metric for real-world application (i.e. pinpointing a region for potential public health interventions), the fraction of cases contained within the top percentages of predicted probability-ranked mesh blocks was calculated. Here, the mesh blocks were ordered according to decreasing predicted class probability, with the fraction of total cases present in the top 5%, 10%, 20%, and 50% of ranked mesh blocks recorded.

Due to the existence of many mesh block probability values for the null models having equal values (mostly zero), a randomization approach was employed to eliminate sorting artifacts. Here, a random seed was used 100 times to add a column of random integers (range 1–100) to the data frames containing model predictions. This data column was then used to initially sort the matrix prior to sorting on the predicted probability values. After 100 sorts on the randomized column followed by sorting on the probability values, the order of each mesh block was recorded and used to calculate an average order value. A final sorting operation was performed on the average order value to determine the mesh blocks that were in the top percentages.

## Results

### Possum excreta sample collection and IS*2404* real-time PCR testing

Standardized, grid-sectored roadside collection of possum excreta at approximately 200 m intervals was performed in all freely accessible residential areas of the western tip of the Mornington Peninsula

**Table 1.** Overview of the Mornington Peninsula excreta surveys across two seasons and *M. ulcerans* IS*2404* PCR screening results.

| Season | Possum species | No. of samples positive | No. of samples tested | Positivity rate | Sites with no excreta found |
|---|---|---|---|---|---|
| Summer | | 111 | 987 | 11% | 58 |
| | Ringtail possum | 110 | 947 | 13% | |
| | Brushtail possum | 1 | 40 | 3% | |
| Winter | | 199 | 1,295 | 15% | 62 |
| | Ringtail possum | 192 | 1,237 | 16% | |
| | Brushtail possum | 7 | 58 | 12% | |

(*Figure 1—figure supplement 1*). A total of 2,402 locations were visited during the two sampling seasons (summer: November 2018 - February 2019 and winter: May 2019 – August 2019). We encountered copious amounts of possum excreta during the surveys. In only 120 of all visited locations (5%) no excreta was found within the allotted 5 min survey time per site. This observation indicates that the residential areas of the Mornington Peninsula support a large possum population.

During the surveys, excreta from common ringtail and common brushtail possums (hereafter referred to as ringtail and brushtail possums) were identified. Brushtail possum excreta was less frequent than ringtail excreta. *M. ulcerans* DNA was detected by IS*2404* real-time PCR in 310 of all 2,282 (13.6%) excreta specimens (*Table 1*). A real-time PCR cycle threshold (Ct) value $\leq$40 was considered positive for *M. ulcerans*. This threshold was based on previous performance evaluations of the assay (*Wallace et al., 2017*). The *M. ulcerans* burden within the possum excreta was estimated by reference to an IS*2404* real-time PCR standard curve as described previously (*Wallace et al., 2017*).

All extraction negative controls showed the expected results, indicating that no contamination was introduced during the DNA extraction process. Similarly, all real-time PCR controls (positive and negative controls) produced the expected results, indicating that the amplification reaction was specific and sensitive. Furthermore, all IPCs were detected in the samples, demonstrating the efficiency of the real-time PCR reaction and the absence of inhibitory substances. Overall, the successful performance of the extraction and real-time PCR controls, as well as the internal positive controls, provides confidence in the accuracy and reliability of the molecular detection assay used in this study.

## Impact of season on *M. ulcerans* presence in possum excreta

In southeast Australia, the majority of *M. ulcerans* transmission occurs in the summer months (*Loftus et al., 2018*; *Trubiano et al., 2013*). We, therefore, tested the hypothesis that significantly more *M. ulcerans*-positive possum excreta material would be collected during summer. However, we observed no significant difference between excreta IS*2404* positivity and the sampling season (p=0.2933, Fisher's exact test). Additionally, no difference was found between the proportion of *M. ulcerans* IS*2404*-positive excreta specimens and possum species (p=0.1309, Fisher's exact test).

## Changes in *M. ulcerans* concentration in possum excreta over time and space

We used previously established IS*2404* real-time PCR standard curves to estimate the *M. ulcerans* load in possum excreta material. The Ct values for IS*2404* real-time PCR ranged from 25.6 to 40.0, corresponding to an estimated *M. ulcerans* excreta load of 5–24,000 mycobacterial genomes per fecal pellet (*Wallace et al., 2017*). Interestingly, we noted that sampling season had a small but statistically significant impact on the fecal mycobacterial load (t(308)=-2.4, p=0.0171). On average, *M. ulcerans* positive excreta analyzed in summer had a Ct(IS*2404*) that was 0.87 lower than excreta collected in winter, a difference which corresponds with a 1.8 times higher mycobacterial load in summer excreta material. We observed no statistically significant difference between the fecal mycobacterial loads of the two possum species (*Figure 1*).

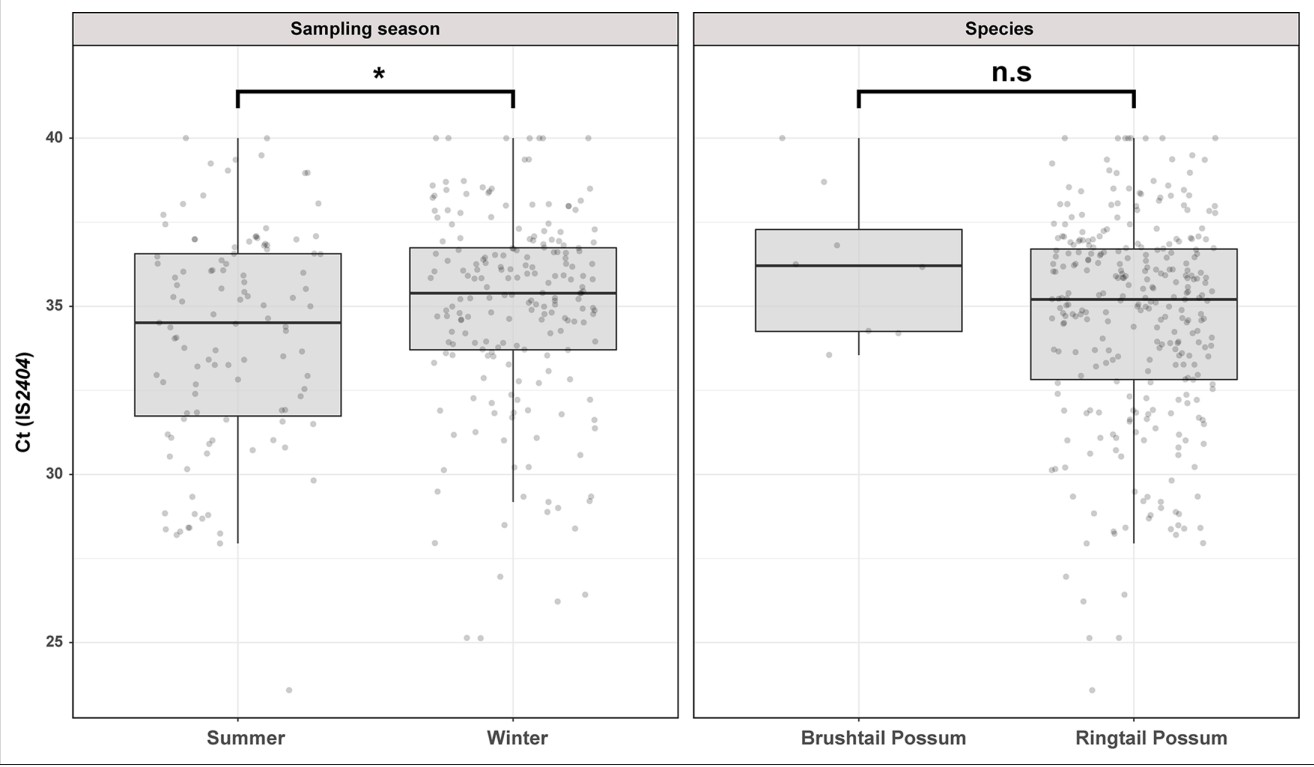

**Figure 1.** Overview of the Mornington Peninsula *M. ulcerans* DNA concentrations in IS*2404* positive excreta stratified by sampling season and possum species. Box plots depict the median, the 25th (lower hinge), and the 75th percentiles (upper hinge) of real-time PCR Ct (IS*2404*) values for each category. The whiskers extend from the Inter-Quartile Range (IQR) hinges to the largest value no further than 1.5 * IQR from the hinges. The null hypothesis (no difference between Ct (IS*2404*) means) was rejected for $p < 0.05$ and assessed using an unpaired t-test while assuming equal variance (checked with an F-test). 'Summer' n=111, 'Winter' n=199, 'Brushtail' n=8, 'Ringtail' n=302'.

The online version of this article includes the following figure supplement(s) for figure 1:

**Figure supplement 1.** Geography of the Mornington Peninsula and Geelong.

**Figure supplement 2.** A set of screenshots demonstrating electronic data collection on the Open Data Kit (ODK) Collect Android app running on a survey phone.

**Figure supplement 3.** Schematic 96-well plate stack of all experimental controls used during 30 separate extraction and real-time PCR runs.

## Spatial distribution of *M. ulcerans*-positive possum excreta across the Mornington Peninsula

The geographical distribution of *M. ulcerans* DNA in the Mornington Peninsula was investigated by mapping all GPS positions of sampling locations visited during the excreta surveys (*Figures 2 and 3*). Across the two sampling seasons, spatial scan statistics revealed three statistically significant geographical areas where IS*2404*-positive excreta clustered non-randomly (*Table 2*). Of note, the Sorrento SaTScan summer cluster encompassed an area where excreta with the highest *M. ulcerans* DNA concentrations of this study was also identified (*Figure 2*). Interestingly, the spatial distribution of IS*2404*-positive possum excreta shifted to some extent between summer and winter sampling, suggesting possum populations might undergo waves of carriage over time (*Figure 3—figure supplement 1*).

## *M. ulcerans*-positive possum excreta to predict occurrence of BU in humans

A major aim of this research was to try and use the possum excreta survey data to predict the risk of *M. ulcerans* transmission to humans. Our approach was to prospectively compare the non-random clusters of *M. ulcerans*-positive possum excreta samples described above with any non-random clusters of human BU cases that were likely acquired during the summer and winter sampling seasons.

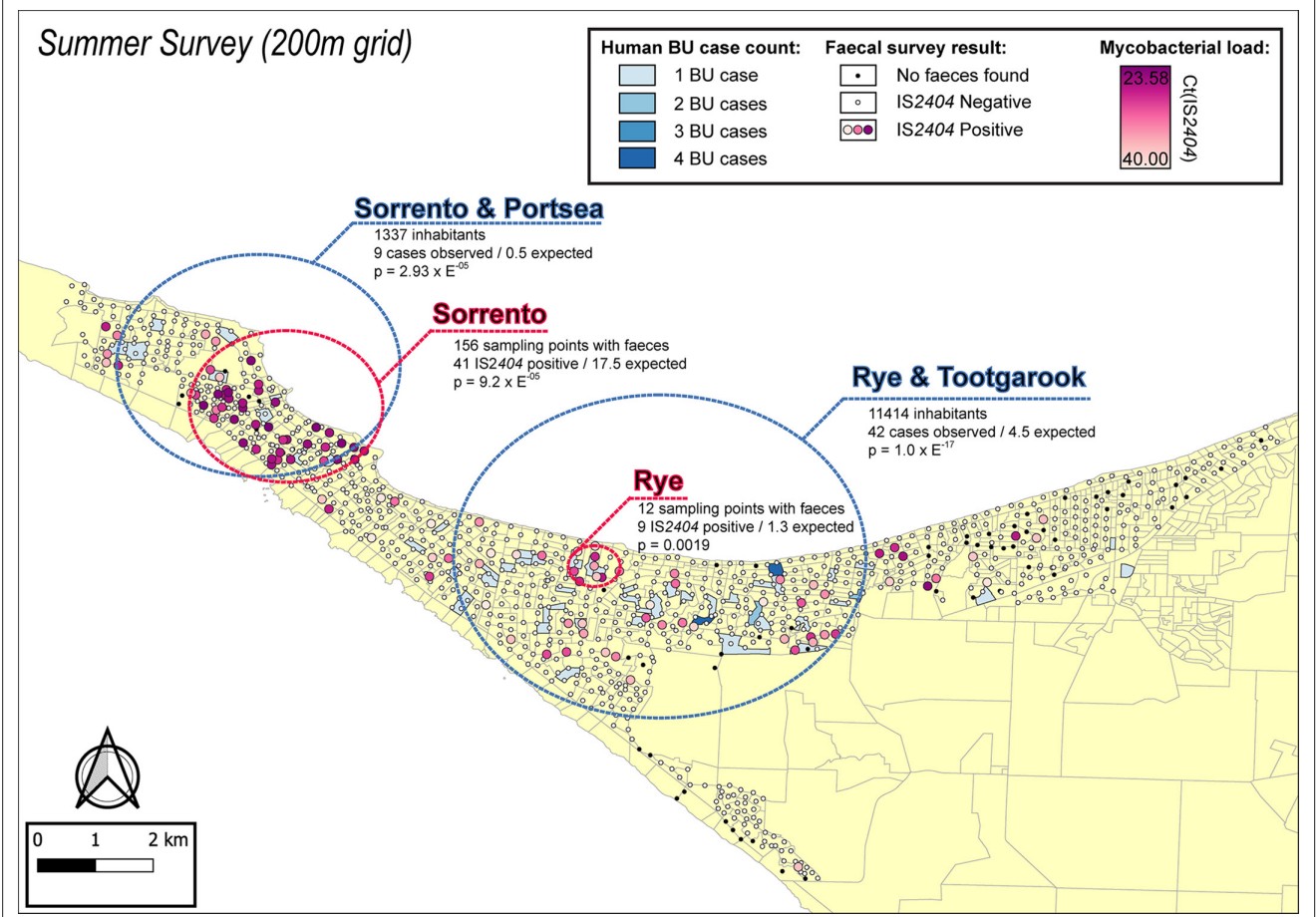

**Figure 2.** Geographical surveillance of *M. ulcerans* in the Mornington Peninsula during the southern hemisphere's summer. The distribution of points where possum excreta was sampled along a 200 m grid pattern is presented alongside with IS*2404* molecular screening results. The pink to purple color gradient visualizes inferred mycobacterial loads in analyzed excreta as estimated from IS*2404* cycle thresholds. The dashed red circles represent significant (p<0.005) non-random clustering of IS*2404* positive possum excreta identified with spatial scan statistics. All Buruli ulcer (BU) patients notified to the Department of Health (DH) with an inferred exposure time that overlapped with the excreta survey organized during summer are tabulated here by mesh block. A gradient is used to illustrate BU case counts per mesh block. The dashed blue circles represent geographical areas with higher rates of BU than would be expected if the risk of contracting BU was evenly distributed across the Peninsula.

The online version of this article includes the following figure supplement(s) for figure 2:

**Figure supplement 1.** Selection of two populations of Buruli ulcer (BU) patients with exposure intervals that aligned with the excreta possum surveys organized during the southern hemispheres' summer and winter.

To do this, a de-identified DH database extract with enhanced surveillance data was used to select BU patients infected with *M. ulcerans* in the Mornington Peninsula during an exposure interval that aligned with the summer and winter excreta possum surveys (*Figure 2—figure supplement 1*). As a result, two populations of BU patients were identified that were highly likely to have been infected in the Mornington Peninsula during the summer (n=62) and winter (n=35) excreta surveys. On the assumption that residents were infected near their houses, the address of the property (geocoded to the 2011 mesh block level) was used in all spatial analyses. Additionally, the population-at-risk from which BU cases arose was represented by the Mornington Peninsula's 144,817 inhabitants as recorded in the 2011 census.

Spatial scan statistics applied to these mesh block data for these BU cases revealed three statistically significant clusters across the two sampling seasons where human BU cases aggregated non-randomly. Across the two sampling seasons, there was a significant spatial correlation between the three clusters of human BU disease and the three clusters with high occurrence of *M. ulcerans* positive possum excreta (*Table 3, Figures 2 and 3*).

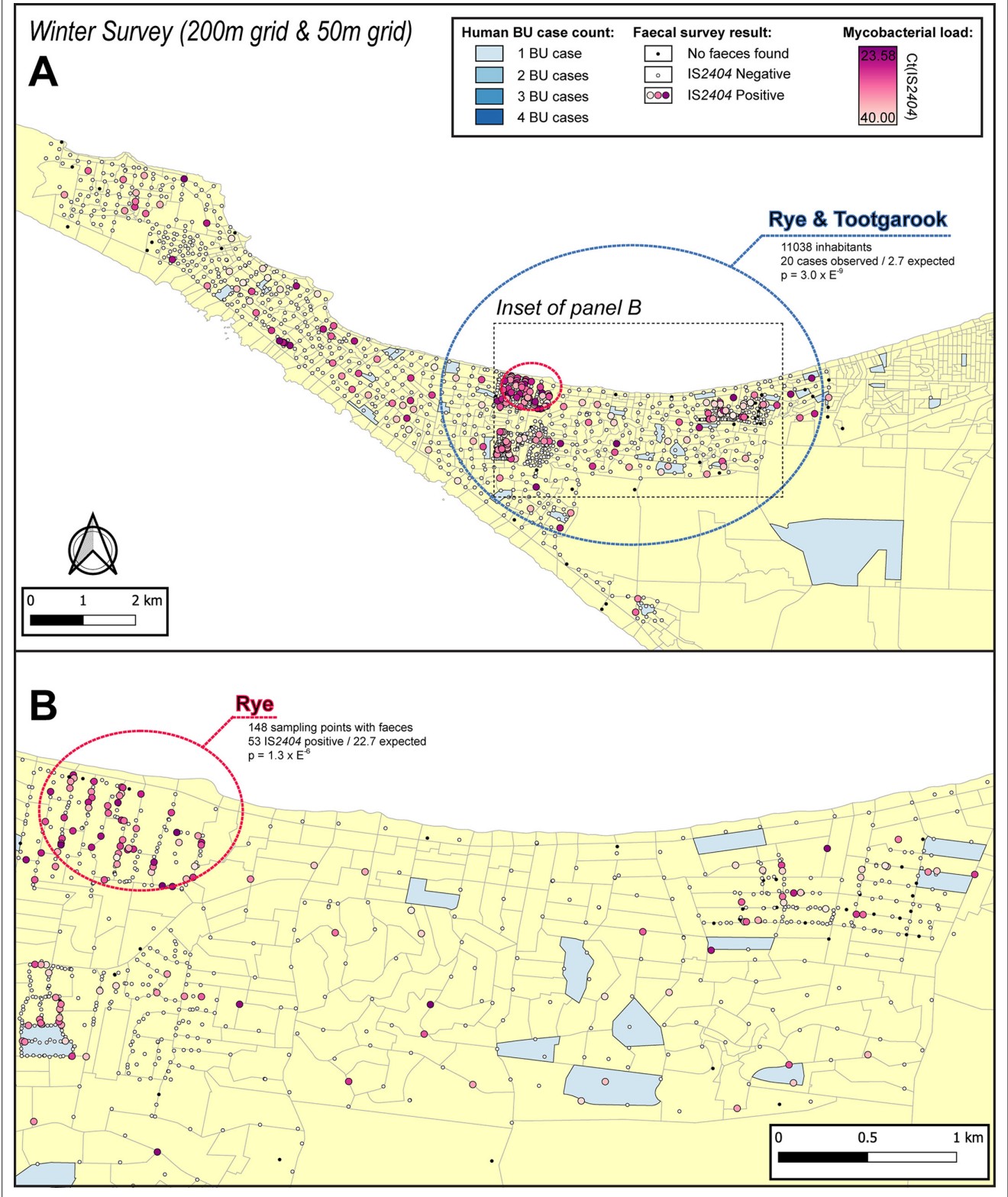

**Figure 3.** Geographical surveillance of *M. ulcerans* in the Mornington Peninsula during the southern hemisphere's winter. The locations where the standardized roadside collection of possum excreta was organized are illustrated alongside with IS*2404* molecular screening results. The winter survey was performed along a 200 m grid pattern although three limited regions were additionally sampled at higher resolution along a 50 m grid (detailed in panel B). All Buruli ulcer (BU) patients notified to the Department of Health (DH) with an inferred exposure time that overlapped with the excreta survey

*Figure 3 continued on next page*

*Figure 3 continued*

organized during winter are tabulated here by mesh block. The dashed circles represent geographical areas with higher rates of IS*2404*-positive possum excreta (red) or BU disease in humans (blue).

The online version of this article includes the following figure supplement(s) for figure 3:

**Figure supplement 1.** Comparative seasonal distribution of *M. ulcerans* in the Mornington Peninsula during the southern hemisphere's summer and winter.

The statistical model developed to prospectively predict the probability of a mesh block containing a case demonstrated a superior ability under spatial-block cross-validation (*Figure 4—figure supplement 1*) to rank mesh blocks according to whether they contained a case or not during the exposure interval – the Receiver-Operating-Characteristic (ROC) AUC value - with a mean AUC of 0.66. This is compared to a mean AUC of 0.56 for a null model based on the preceding year's human BU case incidence data for the Mornington Peninsula (*Figure 4A and B*). When fitted to the full Mornington Peninsula dataset, the model parameters were estimated as: $\beta$ = 0.014 (95% confidence interval 0.01–0.02) and $\sigma$ = 1.06 (0.92–1.21).

For the full out-of-sample validation test, the model developed on the Mornington Peninsula data was validated against possum excreta survey data of 1,128 sites and 661 excreta specimens collected during 2020 in the Geelong region (*Figure 5*), and BU cases in the same region. Three confirmed human BU cases were reported from the Geelong region with a transmission interval that overlapped the sampling period (*Figure 5* and *Figure 5—figure supplement 1*). The model achieved an AUC of 0.75 (*Figure 4C*), compared with a null model based on the previous year BU case incidence with an AUC score of 0.50, underscoring the predictive ability of the excreta-informed model (*Figure 4D*).

In addition to the AUC, we also calculated for all models the fraction of total mesh blocks with cases that were detected when targeting the top 5%, 10%, 20%, and 50% of mesh blocks, ranked by the predicted probability that a mesh block will contain at least one BU case (*Figure 4E*). The excreta-informed models deployed in both the Mornington Peninsula and Geelong areas demonstrated a greater ability to classify case-containing mesh blocks into the 5%, 10%, 20%, and 50% of top-ranking mesh blocks than that of the null models (*Figure 4E*). Targeting of the top 20% of total mesh blocks for these two regions represents a substantial reduction in the overall number of mesh blocks at 368/1,840 and 365/1,827 for the Mornington Peninsula and Geelong models, respectively. Given the tradeoff between narrowing the geographic search area and maintaining sufficient sensitivity to detect mesh blocks where cases might occur, we found that the selection of the top 20% of excreta-informed model probability-ranked mesh blocks was a good compromise. The geographical distribution of the top 20% of probability-ranked mesh blocks obtained with the excreta-informed models also had obvious spatial clustering compared with the null models (*Figure 6A–D*). The non-random mesh block probability density of the excreta-informed models (*Figure 6B and D*) suggests these data can be used to inform rational, targeted, and thus more cost-effective deployment of any interventions compared with reliance on human case data alone.

We also explored the use of a wider exposure period of 61–277 days, which encompasses the range of observed incubation periods (*Loftus et al., 2018*; *Trubiano et al., 2013*). However, this approach had an overall lower predictive capacity compared to the model based on the IQR of incubation periods (101–171 days). We observed a decrease in model AUC and the proportion of case-containing mesh blocks using the wider exposure period, indicating that the IQR is likely a more accurate estimation of the incubation period (*Supplementary file 1*).

**Table 2.** Details of geographical clusters with high rates of IS*2404*-positive possum excreta identified in SaTScan. LLR = Log-Likelihood Ratio.

| Survey | Approx. location | Cluster radius | # sampling locations | # IS*2404* POS samples observed | # IS*2404* POS samples expected | LLR | p-value |
|---|---|---|---|---|---|---|---|
| Summer | Sorrento | 1.7 km | 156 | 41 | 17.5 | 17.052 | 9.20E-05 |
| Summer | Rye | 0.4 km | 13 | 9 | 1.3 | 13.583 | 1.90E-03 |
| Winter | Rye | 0.6 km | 148 | 53 | 22.7 | 21.800 | 1.27E-06 |

**Table 3.** Details of geographical clusters with high Buruli ulcer (BU) incidence identified in SaTScan.
MB = mesh block, LLR = Log-Likelihood Ratio.

| Survey | Approx. location | Cluster radius | # MB centroids | Cluster census population | BU cases observed | BU cases expected | LLR | p-value |
|---|---|---|---|---|---|---|---|---|
| Summer | Sorrento & Portsea | 2.4 km | 140 | 1,337 | 9 | 0.5 | 17.749 | 2.94E-05 |
| Summer | Rye & Tootgarook | 3.6 km | 408 | 11,414 | 42 | 4.5 | 75.087 | 1.00E-17 |
| Winter | Rye & Tootgarook | 3.6 km | 383 | 11,038 | 20 | 2.7 | 28.770 | 3.05E-09 |

## Discussion

The rapid expansion of BU endemic areas in southeastern Australia has highlighted the need for new strategies to understand and control disease spread. The long and variable incubation period of BU in humans has challenged traditional epidemiological surveillance approaches in tracking the emergence and movement of BU disease foci (*Loftus et al., 2018*; *Trubiano et al., 2013*). Building on the findings of other zoonotic pathogen surveillance programs (*Hamer et al., 2012*; *Childs et al., 2007*), here, we continued to explore the potential role of arboreal marsupial mammals as wildlife sentinels to monitor BU emergence and spread and to help understand the role of these animals in the transmission of BU. Our primary goal was to determine whether *M. ulcerans* surveillance of possum excreta could act as an early warning system capable of predicting future human BU case locations. As a first task, we established a possum excreta surveillance program that monitored *M. ulcerans* in the environment of the Mornington Peninsula. This allowed us to determine the extent of epizootic activity during consecutive summer and winter seasons.

We identified a significant spatial correlation between clusters of *M. ulcerans* positive possum excreta and clusters of confirmed human BU cases likely infected with *M. ulcerans* in the same region during an exposure interval that aligned with the excreta possum surveys (*Figures 2 and 3*). While the overlap between the two cluster types was not perfect, it is important to highlight that the *SaTScan* clusters detected represent the general area of a cluster and the circles are only approximate boundaries. Importantly however, the patterns and overlap we observed aligned with previous assessments of the positive association between *M. ulcerans* in possums and human BU cases (*Fyfe et al., 2010*; *Carson et al., 2014*; *Blasdell et al., 2022*) and thus very strongly implicate Australian native possums as key environmental reservoirs of *M. ulcerans*, involved in a transmission cycle with humans. Of note too, the frequency of excreta positivity for *M. ulcerans* we observed (13.6% for common ringtail possums) was similar to an earlier, smaller survey from the Mornington Peninsula (9.3%) (*Fyfe et al., 2010*). The difference in the seasonal distribution of IS*2404*-positive possum excreta (*Figure 3— figure supplement 1*) point to the importance of conducting these excreta surveys during the warmer summer months when *M. ulcerans* transmission risk is highest (*Loftus et al., 2018*; *Trubiano et al., 2013*). This association between possums, humans, and BU flags this disease as a One Health issue. It also suggests that a surveillance program that monitors *M. ulcerans* DNA in possum excreta could alert public health authorities to increased human BU risks, which would allow prevention and control programs to be implemented before human BU cases occur.

To further explore the potential of possum excreta surveys to predict the risk of BU cases occurring in humans in particular areas, we built a custom statistical model and compared its performance to null models built from the previous year's human BU case incidence. From a public health perspective, the null models can be considered as a conventional approach to determine where cases might appear in the future. The development of an excreta-informed model for the Mornington Peninsula data revealed that it had greater predictive capacity than the Mornington Peninsula null model, in that it could more accurately predict areas (mesh blocks) with cases than the null model. We extended the Mornington Peninsula excreta-informed model to make predictions upon a previously unseen excreta survey dataset in the Geelong region. As for predictions made on the Mornington Peninsula, the predictions made with the Geelong excreta data had a greater ability to correctly predict human BU case-containing mesh blocks than was observed with a null model that used the previous year's BU incidence.

This increased performance of the excreta-informed models might be explained by several factors. First, the long incubation period makes it difficult to establish both when and where a person may

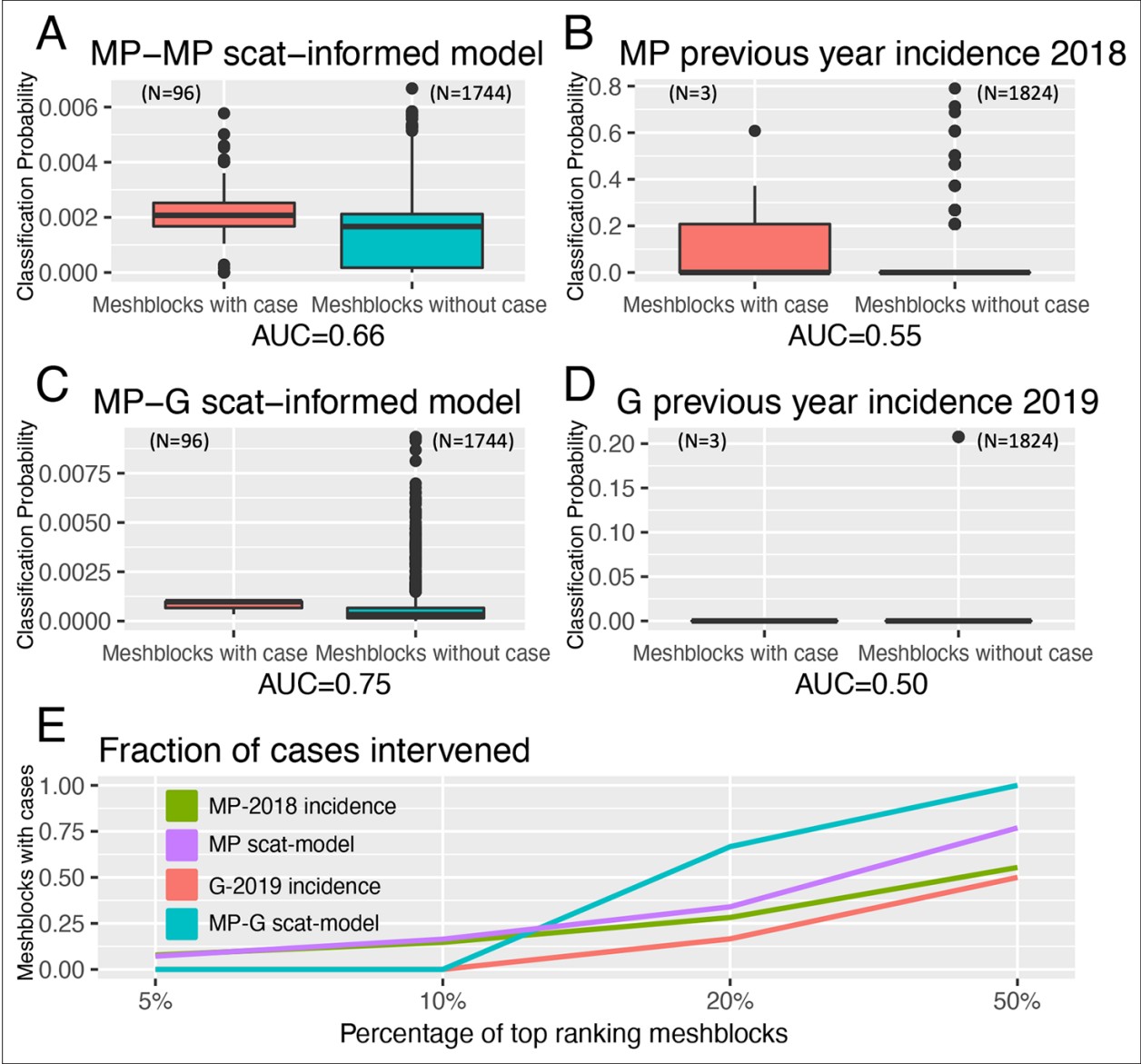

**Figure 4.** Statistical modeling approaches to prospectively predict the likelihood that a mesh block will contain a Buruli ulcer (BU) case. Mornington Peninsula and Geelong have been abbreviated as (MP) and (G), respectively. All paired boxplots show the predicted probabilities for mesh blocks with a case (red) and mesh blocks without a case (blue) and the area-under-the-curve (AUC) value below each graph. (**A**) Paired boxplot of the Mornington Peninsula excreta-informed model. (**B**) Paired boxplot of the Mornington Peninsula excreta-informed model when predicting on the Geelong data. (**C**) Paired boxplot of the Mornington Peninsula previous year's incidence (2018) null model. (**D**) Paired boxplot of the Geelong previous year's incidence (2019) null model. All paired boxplots show the median and interquartile range (**E**) Ranked performance of all predictive models. Ranking cutoff intervals included the top 5%, 10%, 20%, and 50% of mesh blocks, ordered by their declining predicted class probabilities.

The online version of this article includes the following figure supplement(s) for figure 4:

**Figure supplement 1.** Spatial block cross-validation approach.

have been infected with *M. ulcerans*, so the spatial information used by the null models will likely contain more 'noise' than the excreta-informed models. Possums, however, have a limited range, usually less than 100 m, and so their excreta is a spatially trustworthy analyte, providing a more accurate picture of pathogen distribution in the environment (*Fyfe et al., 2010*; *Buultjens et al., 2017*). Second, human BU cases are not detected in areas where possums do not harbor *M. ulcerans* (*Fyfe et al., 2010*; *Carson et al., 2014*; *Blasdell et al., 2022*). Therefore, possums are playing a substantially more contributive role than humans to *M. ulcerans* transmission cycles, helping to explain why the excreta-informed model outperforms human BU case, incidence-based models. The ability of

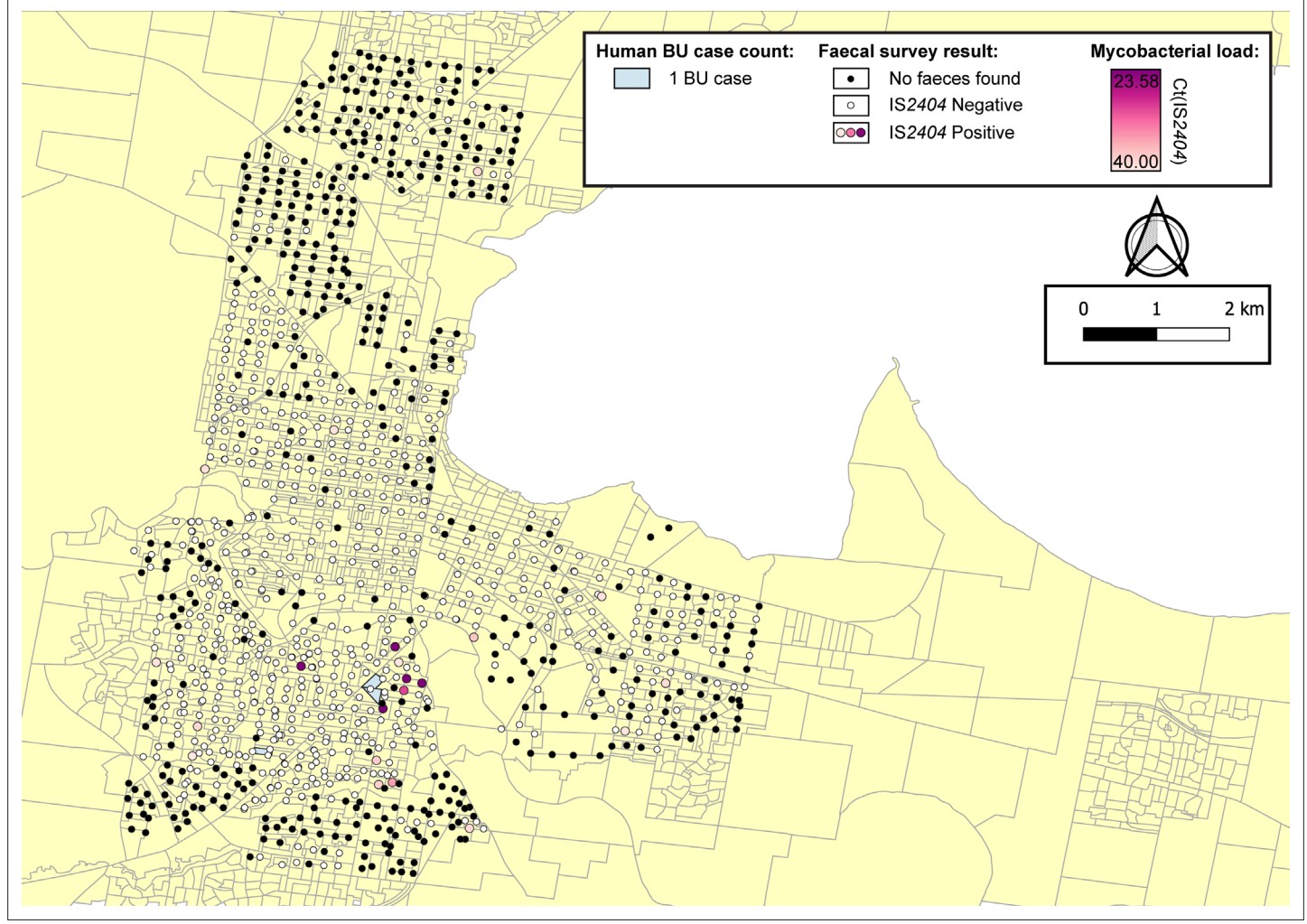

**Figure 5.** Geographical surveillance of *M. ulcerans* in the Geelong region. The distribution of points where possum excreta was sampled along a 200 m grid pattern is presented alongside with IS*2404* molecular screening results. The pink to purple color gradient visualizes inferred mycobacterial loads in analyzed excreta as estimated from IS*2404* real-time PCR results. All BU patients were notified to the Department of Health (DH) with an inferred exposure time that overlapped with the Geelong excreta survey organized between January 16, 2020 and April 28, 2020 tabulated here by mesh block. *M. ulcerans* DNA was detected by IS*2404* real-time PCR in 21 of the 661 (3.2%) excreta specimens (*Table 4*). As observed with the Mornington Peninsula survey, Ringtail possum excreta was more frequently found than Brushtail possum excreta but the proportion IS*2404*-positive was not significantly different (p=1, Fisher's exact test) (*Table 4*).

The online version of this article includes the following figure supplement(s) for figure 5:

**Figure supplement 1.** Population of Buruli ulcer (BU) patients with exposure intervals that aligned with the Geelong excreta possum survey organized between January 16 and April 28, 2020.

**Table 4.** Geelong excreta survey results summary.

| Dates | Possum species | No. of samples positive | No. of samples tested | Positivity rate | Sites with no excreta found |
|---|---|---|---|---|---|
| | | 21 | 661 | 3% | 467 |
| | Ringtail possum | 19 | 577 | 3% | |
| January- April 2020 | Brushtail possum | 2 | 84 | 2% | |

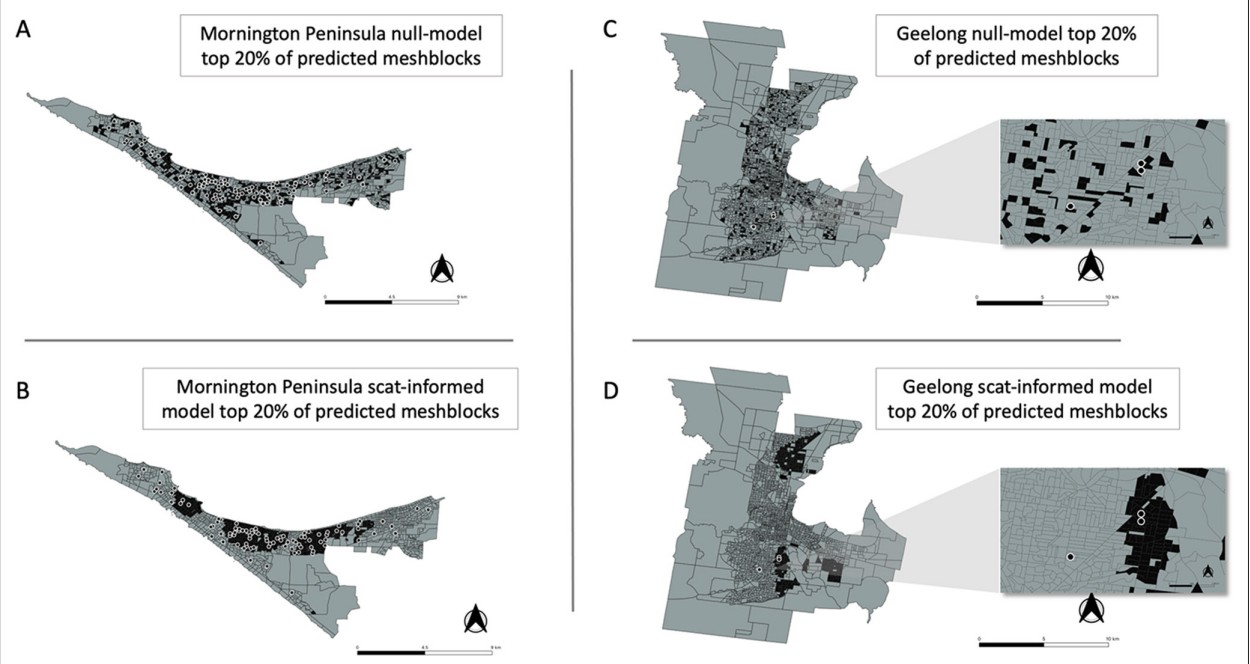

**Figure 6.** Geographical distribution of the top 20% of probability-ranked mesh blocks for all excreta-informed and null models. Black indicates mesh blocks in the top 20% of probability-ranked results while grey indicates mesh blocks in the bottom 80%. Black circles with white borders indicate location of confirmed BU cases that occurred within the transmission window following the excreta sampling. (**A**) Mornington Peninsula null-model top 20% mesh block predictions. (**B**) Mornington Peninsula excreta-informed model top 20% mesh block predictions. (**C**) Geelong null-model top 20% mesh block predictions. (**D**) Geelong excreta-informed model top 20% mesh block predictions. Insets show zoom-in of mesh blocks where Buruli ulcer (BU) cases occurred.

the excreta-informed model trained with data from the Mornington Peninsula to predict human case occurrence in a distinct area (Geelong) strongly reinforces the link between possums harboring *M. ulcerans* and human BU cases across different geographic areas in southeast Australia.

The ranking evaluation of the predictive models is another strength of the modeling approach, as it reports the number of mesh blocks where cases potentially could have been prevented (or better managed) if these top-ranking mesh blocks were targeted by an effective public health intervention. We envisage a scenario in which such geographical risk assessments could form the basis of public health messaging programs that target areas where disease transmission is predicted as most likely to occur. In this way, for instance, frontline general practice clinicians could be advised of the elevated local *M. ulcerans* transmission risk, potentially leading to earlier patient diagnosis and improved clinical outcomes for the cases that do emerge. In addition, targeted messaging based on predictive modeling may also encourage preventative behaviors among local communities that could lessen the chances of transmission and reduce the number of emerging cases. For example, targeted messaging promoting behaviors that mitigate known BU risk factors (e.g. mosquito bites) or promote known protective behaviors (use of insect repellent) could reduce disease incidence (**Blasdell et al., 2022**; **Quek et al., 2007**). Other interventions could reduce the abundance of potential vectors that might be transmitting *M. ulcerans* from possums to people, or perhaps seek to control the infection in possums with novel therapeutic interventions.

In addition to using the IQR of observed incubation periods to define the exposure period, we also investigated the effect on our model of the use of a wider window using the absolute range of observed incubation periods. While the absolute range of observed incubation periods provides a comprehensive understanding of the spread of incubation periods, we found that modeling based on the incubation period IQR provided better predictive capacity. This suggests that the IQR better captures the central tendency of the true incubation period, allowing for the building of better predictive models. Therefore, we recommend using the IQR as a summary measure of the incubation period when conducting future studies or developing models to predict the spatial distribution of BU cases. Other opportunities to refine and improve the model might be possible by improving the diagnostic

yield from the possum excreta. It may be that only reporting excreta that has viable *M. ulcerans* would best inform our predictive risk models. The challenges of bacteriological culture for *M. ulcerans* from microbially complex specimens make this viability test impractical and the IS*2404* real-time PCR test gives no indication of bacterial viability. However, a recently described RNA-based *M. ulcerans* PCR test used for possum excreta might be an alternative assay for inferring *M. ulcerans* viability (*Blasdell et al., 2022*).

Possum excreta fecal surveys are practical and cost-effective because excreta from Australian native possums in urban and semi-urban areas is highly abundant, easily recognized, and easily accessed. It is an ideal environmental analyte. Roadside collection of possum excreta and subsequent molecular screening is also relatively straightforward and processing epizootiological data does not require informed consent or access to medical records. Furthermore, as multiple years may elapse between the occurrences of BU in a particular region, continuous excreta surveillance might detect trends in the distribution and epidemiology of BU in a region and allow the effectiveness of any BU control measures; measures such as identifying and treating *M. ulcerans*-infected possum populations.

We explored the impact of sampling at a higher density (50 m intervals instead of 200 m) (*Figure 3A*). The higher density sampling provided by 50 m intervals increased resolution for the fecal mapping, but it didn't materially change the pattern of *M. ulcerans* positive fecal samples detected at 200 m sampling grids. We propose that 200 m sampling grids provide a pragmatic balance between survey sensitivity and survey time/costs.

The streamlined workflow, the custom-built distance trackers, and the use of Android mobile phones equipped with an electronic data collection solution simplified the fieldwork to such an extent that new field workers could be trained within a day. Moreover, incorporating ODK in our workflow allowed us to organize our mobile excreta surveys in a cost-effective manner as the open-source software suite was hosted free of charge by a cloud provider. Furthermore, ODK Collect proved a powerful tool for ensuring high-quality data collection as it only uploaded completed submissions, as the app automatically validated survey responses at the point of data collection by using entry constraints, error checks, and form logic. This significantly reduced data errors and data loss which would be much more common in paper surveys at this scale.

In this study, we provide further evidence that BU in southeast Australia is a zoonotic infection involving Australian native possums in a transmission cycle. The means by which the pathogen is spread between possums and humans is under active investigation, but mosquitoes are likely vectors (*Johnson et al., 2007*; *Lavender et al., 2011*). We are also investigating the natural history of *M. ulcerans* infection in possums, to better understand the apparent susceptibility of these animals to mycobacteria. The detection of *M. ulcerans* DNA in possum excreta is associated with the occurrence of BU disease in humans and we have explored how these surveillance data can be used to predict future BU case emergence. A future surveillance program should collect, analyze, and interpret epidemiological, clinical, and epizootiological data on BU. Questions/issues to address about such a program include the breadth of areas to survey, frequency of specimen collection, and availability of the resources/trained personnel required to establish and maintain a program. Environmental surveillance should identify epizootics as quickly as possible so that steps can be taken to control disease spread. We have found possum excreta surveillance of *M. ulcerans* DNA can identify the spread of BU epizootics and provide public health authorities with sufficient warning to implement control measures before human cases occur. Finally, whilst this environmental surveillance system will assist BU control in southwest Australia, it will not directly translate to other BU endemic areas of the world, particularly the high-burden regions of West Africa, where there are no possums, and an equivalent animal reservoir is yet to be found.

## Acknowledgements

We are grateful to Gabrielle Stinear, Andrew Walker, Brianna Behrsin, Zoe Winkle, Zoe James, Simone Clayton, Kerri Howell, Jake A Linke and the Environmental Health team of the Mornington Peninsula Shire for valued assistance during fieldwork.

## Additional information

### Funding

| Funder | Grant reference number | Author |
|---|---|---|
| National Health and Medical Research Council | GNT1152807 | Timothy P Stinear |
| National Health and Medical Research Council | GNT1196396 | Timothy P Stinear |

The funders had no role in study design, data collection and interpretation, or the decision to submit the work for publication.

### Author contributions

Koen Vandelannoote, Conceptualization, Data curation, Software, Formal analysis, Validation, Investigation, Visualization, Methodology, Writing – original draft, Project administration, Writing – review and editing; Andrew H Buultjens, Data curation, Software, Formal analysis, Validation, Investigation, Visualization, Methodology, Writing – original draft, Writing – review and editing; Jessica L Porter, Formal analysis, Investigation, Methodology, Writing – original draft; Anita Velink, Investigation, Methodology; John R Wallace, Conceptualization, Investigation, Methodology, Writing – review and editing; Kim R Blasdell, Resources, Supervision, Investigation, Methodology, Writing – review and editing; Michael Dunn, Investigation, Project administration; Victoria Boyd, Data curation, Investigation, Methodology; Janet AM Fyfe, Conceptualization, Data curation, Investigation, Methodology; Ee Laine Tay, Investigation, Methodology, Writing – review and editing; Paul DR Johnson, Conceptualization, Supervision, Funding acquisition, Investigation, Methodology, Writing – original draft, Project administration, Writing – review and editing; Saras M Windecker, Conceptualization, Software, Formal analysis, Investigation, Methodology, Writing – review and editing; Nick Golding, Conceptualization, Data curation, Software, Formal analysis, Supervision, Funding acquisition, Validation, Investigation, Methodology, Writing – original draft, Writing – review and editing; Timothy P Stinear, Conceptualization, Data curation, Formal analysis, Supervision, Funding acquisition, Validation, Investigation, Visualization, Methodology, Writing – original draft, Project administration, Writing – review and editing

### Author ORCIDs

Koen Vandelannoote (iD) http://orcid.org/0000-0001-8367-4083
Andrew H Buultjens (iD) http://orcid.org/0000-0002-5984-1328
Saras M Windecker (iD) http://orcid.org/0000-0002-4870-8353
Nick Golding (iD) http://orcid.org/0000-0001-8916-5570
Timothy P Stinear (iD) http://orcid.org/0000-0003-0150-123X

### Ethics

Ethical approval for the use in this study of de-identified human BU case location, aggregated at mesh block level, was obtained from the Victorian Government Department of Health Human Ethics Committee under HREC/54166/DHHS-2019-179235(v3), 'Spatial risk map of Buruli ulcer infection in Victoria.'.

### Decision letter and Author response

Decision letter https://doi.org/10.7554/eLife.84983.sa1
Author response https://doi.org/10.7554/eLife.84983.sa2

## Additional files

### Supplementary files
• MDAR checklist

• Supplementary file 1. Impact on model performance by expanding the exposure window.

### Data availability

The computer code and source data used in this study are available here: https://github.com/abuultjens/Possum_scat_survey_predict_human_BU, (copy archived at *Buultjens, 2023*).

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
