## [Editor Report]

This study is an important contribution to the understanding of Buruli ulcer transmission in Australia. The authors provide compelling evidence that the carriage of Mycobacterium ulcerans by possums, within their small home range, can predict cases of Buruli ulcer disease in individuals who visit those areas. While not directly relevant to the transmission of Buruli ulcer in West and Central Africa, the work will be of great interest to those studying the transmission of opportunistic environmental pathogens.

---

## [Decision Letter]

**Decision letter after peer review:**

Thank you for submitting your article "Statistical modelling based on structured surveys of Australian native possum excreta harbouring *Mycobacterium ulcerans* predicts Buruli ulcer occurrence in humans" for consideration by *eLife*. Your article has been reviewed by 3 peer reviewers, and the evaluation has been overseen by a Reviewing Editor and Bavesh Kana as the Senior Editor. The reviewers have opted to remain anonymous.

Essential Revisions

Major concerns:

1. In this study, the detection of M. ulcerans DNA in environmental samples was performed by targeting IS2404 by qPCR, and a sample was considered positive if Ct<40. However, it is usual to confirm the presence of M. ulcerans in the environment by the detection of 2 sequences (IS2404 and KR) and to consider a positive sample with an adequate deltaCt between the two sequences and Ct<35 cycles. Furthermore, since the authors found a correlation between bacterial estimated load in faeces and human UB focus, it would be all the more judicious to fix an adequate threshold of environmental positivity in their model. It would have been even stronger. Can the authors comment?

2. line 592-593: authors can not claim this confirmation unless they have genotyped and compared human strains and environmental DNA.

3. In their model, the authors have used an assumed "exposure window" for when patients were infected with M. ulcerans in the Mornington Peninsula. Correctly defining, and assigning, this is absolutely critical to the accuracy of the statistical model, as is "blinding" of researchers assigning mesh boxes to patients to the results of surveillance data (and vice versa). These aspects are not fully clear in the current version. Furthermore, the effects on the model of changing these assumptions are not discussed.

4. Comparing the summer and winter surveys at the Mornington Peninsula, the distribution of M. ulcerans positive excreta appears to have changed quite substantially, especially given that the possums are reported to be highly territorial with a range of only 100m. This version of the manuscript does not formally compare these spatial distributions, only the averages. Such an analysis would help understand if it is the possums that are moving, whether the possums undergo 'waves' of carriage (or indeed any other explanation), or if these apparent differences are down to chance.

5. Line 75. "Gut carriage" of Mu by possums is assumed but has never been strictly proven. Successful culture of M. ulcerans from a possum or its excreta has yet to be achieved. While this is likely a technical limitation due to the bacteria's extremely slow doubling time, it should be transparently discussed for the benefit of the wider readership audience of *eLife*.

6. Line 216. More information about how the clinical information was geocoded is required, especially where cases were non-resident. Does the clinical information include information on dates and locations of visits to the area? This isn't clear. Please also provide additional information about how the researchers were blinded to the clinical and surveillance data and at what point they were unblinded to each other's findings.

7. Line 221: Inferring from Figures S4 and S6, it appears that the exposure window is defined as the quoted IQR from ref [4] of 101-171 days. If this is correct, please state it clearly, and expand more on how these values were arrived at. Is it possible to test the effect on the statistical model if this window is adjusted? A window that increases the model's predictive power might be informative with regard to the (still open) question of the BU incubation period.

8. Line 226. Please confirm that for all patients, it is recorded/known that patients had visited the mesh box assigned to them in the geocode within the exposure period defined in Figure S4, and clearly explain how any instances of patients vising more than one mesh box (if this occurred) was handled in the model.

Editorial/presentation of data:

*eLife* recommends a particular colour palette to be fully accessible (Colour Universal Design https://jfly.uni-koeln.de/color/). Several of the figures are quite hard to see even for a non-colourblind person, given the amount of information packed into them. Giving some thought to the best colour scheme would enhance the overall readability of the manuscript.Including some subheadings in the results would also enhance the overall readability.

Figure S1. Please indicate Belmont in panel E.

Figure S3. This legend is inadequate. It should be possible to understand what the figure shows without referring to the text. Presumably, it is the incidence of BU, but more details are needed, and this should be clearly described.

Line 354. Comment on how the positivity rate for Mu compares to other prior possum excreta surveys in the area.

Figure 1. Rather a lot of samples have a Ct above 37. Have the authors tested in the model how changes in the PCR cut-off impact the prediction ability? This legend is also inadequate, as it appears to be solely a title. Ct value is not DNA concentration. The statistical test is not mentioned, please include this and confirm that it can apply to log2 data. Genocopies per sample might be a better metric here.

Figures 2 and 3. Very hard to visualise the mesh blocks in this figure due to the similar colour of the background to the BU case scale (see my general comment above).

Figure S4 and S6 would sit better with the methodology, as it is highly relevant to the section lines 221-228.

Figure 5. Blue/red horribly unclear at the resolution provided. A better colour choice is needed.

Line 519. This would be a good place to briefly acknowledge the limitations of the PCR approach used.

Line 530. Good point to comment on differences in the spatial distribution in the summer and winter surveys.

Discussion point: Since the mosquito seems to be an important mechanical vector for transporting the Mu from possum excreta into humans, their role in this seems currently under-discussed. In particular, what is the range of a mosquito? Is this known? Could this influence the findings?

Discussion point: It is important to include a qualifying statement that the model developed is only applicable to Australia, since no link is currently known between animal excreta or mosquitos in other countries, particularly West Africa where the number of BU cases is highest.

Replace BU transmission with M. ulcerans transmission in the manuscript. BU is the resulting disease.

Abstract /elsewhere: Is really Buruli ulcer a "neglected disease"? Is it really contributive to "One-Health" in that topic?

Abstract, line 37: Clarify the temporality of Buruli ulcer cases/faeces investigation campaign.

Line 63: Were they microbiologically documented cases?

Line 140-174: Any negative control faeces (from animal demonstrated as feces-negative in previous, referenced studies?).

Line 158-160: Was morphological identification, previously validated? Or would genetic identification be an added value?

Line 188-189: Clarify the number of positive/negative controls per experimental batch. Clarify the M. ulcerans- specificity of IS2404.

Line 213: "… detected laboratories…": please correct.

Line 345-357: Give results for negative controls. Give results for positive controls, clarifying the interest of such positive controls for this study.

Line 526: The authors may want to expand this point relative to "reservoir" (although it was not in the spectrum of the study), rather than "source" of infection for the population.

Authors may want to clarify the specific contribution of each author.

*Reviewer #1 (Recommendations for the authors):*

In this study, the detection of M. ulcerans DNA in environmental samples was performed by targeting IS2404 by qPCR and a sample was considered positive if Ct<40. However, it is usual to confirm the presence of M. ulcerans in the environment by the detection of 2 sequences (IS2404 and KR) and to consider a positive sample with an adequate deltaCt between the two sequences and Ct<35 cycles.

Furthermore, since the authors found a correlation between bacterial estimated load in feces and human UB focus, it would be all the more judicious to fix an adequate threshold of environmental positivity in their model. It would have been even stronger.

line 592-593: authors can not claim this confirmation unless they have genotyped and compared human strains and environmental dna.

Replace BU transmission with M. ulcerans transmission in the manuscript.

*Reviewer #2 (Recommendations for the authors):*

This was an extremely interesting paper, which was accessible for this non-statistician, and the weaknesses laid out in the Public Review I'm sure can be addressed straightforwardly. The following detailed comments should help with this:

General: *eLife* recommends a particular colour palette to be fully accessible (Colour Universal Design https://jfly.uni-koeln.de/color/). Several of your figures are quite hard to see even for a non-colourblind person, given the amount of information packed into them. Giving some thought to the best colour scheme would enhance the overall readability of the manuscript.

General: Including some subheadings in the results would also enhance the overall readability.

Line 75. "Gut carriage" of Mu by possums is assumed but has never been strictly proven. I am sure, you are highly aware that the successful culture of M. ulcerans from a possum or its excreta has yet to be achieved. While this is likely a technical limitation due to the bacteria's extremely slow doubling time, it should be transparently discussed for the benefit of the wider readership audience of *eLife*.

Figure S1. Please indicate Belmont in panel E.

Line 188-189. There is little justification given for choosing solely IS2404 as the test of Mu positivity. Please expand this section, and discuss the potential for false positive (other MPM) and false negative, as well as state what the Ct value cut off. Information on the performance of extraction negative controls (in addition to nuclease-free water) should be given to reassure that your processes eliminate cross-contamination.

Line 216. More information about how the clinical information was geocoded is required, especially where cases were non-resident. Does the clinical information include information on dates and locations of visits to the area? This isn't clear. Please also provide additional information about how the researchers were blinded to the clinical and surveillance data and at what point they were unblinded to each other's findings.

Line 221: Inferring from Figures S4 and S6, it appears that the exposure window is defined as the quoted IQR from ref [4] of 101-171 days. If this is correct, please state it clearly, and expand more on how these values were arrived at. Is it possible to test the effect on the statistical model if this window is adjusted? A window that increases the model's predictive power might be informative with regard to the (still open) question of the BU incubation period.

Line 226. Please confirm that for all patients, it is recorded/known that patients had visited the mesh box assigned to them in the geocode within the exposure period defined in Figure S4, and clearly explain how any instances of patients vising more than one mesh box (if this occurred) was handled in the model.

Figure S3. This legend is inadequate. It should be possible to understand what the figure shows without referring to the text. Presumably, it is the incidence of BU, but more details are needed, and this should be clearly described.

Line 354. Comment on how the positivity rate for Mu compares to other prior possum excreta surveys in the area.

Figure 1. See the comment above, rather a lot of samples have a Ct above 37. Have you tested in your model how changes in the PCR cut-off impact the prediction ability? This legend is also inadequate, as it appears to be solely a title. Ct value is not DNA concentration. The statistical test is not mentioned, please include this and confirm that it can apply to log2 data. Genocopies per sample might be a better metric here.

Figures 2 and 3. Very hard to visualise the mesh blocks in this figure due to the similar colour of the background to the BU case scale (see my general comment above).

Figure S4 and S6 would sit better with the methodology, as it is highly relevant to the section lines 221-228.

Figure 5. Blue/red horribly unclear in the resolution I was provided. A better colour choice is needed.

Line 519. This would be a good place to briefly acknowledge the limitations of the PCR approach used.

Line 530. Good point to comment on differences in the spatial distribution in the summer and winter surveys.

Discussion point: Since the mosquito seems to be an important mechanical vector for transporting the Mu from possum excreta into humans, their role in this seems currently under-discussed. In particular, what is the range of a mosquito? Is this known? Could this influence the findings?

Discussion point: It is important to include a qualifying statement that the model developed is only applicable to Australia, since no link is currently known between animal excreta or mosquitos in other countries, particularly West Africa where the number of BU cases is highest.

---

## [Author Response]

Essential RevisionsMajor concerns:1. In this study, the detection of M. ulcerans DNA in environmental samples was performed by targeting IS2404 by qPCR, and a sample was considered positive if Ct<40. However, it is usual to confirm the presence of M. ulcerans in the environment by the detection of 2 sequences (IS2404 and KR) and to consider a positive sample with an adequate deltaCt between the two sequences and Ct<35 cycles. Furthermore, since the authors found a correlation between bacterial estimated load in faeces and human UB focus, it would be all the more judicious to fix an adequate threshold of environmental positivity in their model. It would have been even stronger. Can the authors comment?

Our threshold of Ct <40 is based on the high performance of the IS*2404* Taqman assay: approx. 200 copies of IS*2404* are present in the *M. ulcerans* genome. As a result, the test is highly sensitive and can reliably detect 0.1 – 1 *M. ulcerans* genome equivalents (Ct range 39-40) (manuscript ref. 33). If the concern of reviewers is that the IS*2404* positivity of possum excreta may not reflect the presence of *M. ulcerans*, then we point to our previous research showing that DNA from IS*2404* positive possum excreta can be used for subsequent *M. ulcerans* PCR amplicon SNP typing (DOI: 10.1128/AEM.02612-17). We also highlight our use of randomly interspersed blank DNA extraction controls and no-template PCR controls in every batch of samples tested to guard against PCR contamination (lines 220-232). Regarding the use of IS*2606* and KR-B as confirmatory PCR targets, they are of limited value when *M. ulcerans* abundance is low, as they have reduced sensitivity compared to IS*2404*, so we did not routinely perform these additional tests.

2. line 592-593: authors can not claim this confirmation unless they have genotyped and compared human strains and environmental DNA.

We have previously shown using genomics that *M. ulcerans* infecting possums and humans share the same genotype, as we mention in the introduction (lines 99-100). Our current study further confirms that Buruli ulcer is a zoonosis, by showing a quantitative link between possums shedding *M. ulcerans* and the risk of humans developing Buruli ulcer. Nevertheless, we have qualified the sentence as follows:

“In this study, we provide further evidence that Buruli ulcer in southeast Australia is a zoonotic infection involving Australian native possums in a transmission cycle” (line 768).

3. In their model, the authors have used an assumed "exposure window" for when patients were infected with M. ulcerans in the Mornington Peninsula. Correctly defining, and assigning, this is absolutely critical to the accuracy of the statistical model, as is "blinding" of researchers assigning mesh boxes to patients to the results of surveillance data (and vice versa). These aspects are not fully clear in the current version. Furthermore, the effects on the model of changing these assumptions are not discussed.

The exposure window we use is based on very sound and comprehensive epidemiological data from two studies arriving at the same estimate of the incubation period (manuscript refs. 4,5). To address the reviewer concerns regarding impact of changing this window on model performance we re-ran the model using the absolute range of observed incubation periods (61-277 days). Model performance was slightly worse using the extended range compared to the IQR (101-171 days). We have included this additional analysis in the results. Refer Supplementary File 1 (results: line 649 onwards and discussion: line 730 onwards).

Regarding the important issue of blinding, the mesh block data was loaded into an R script as a numeric ID string and there was no researcher recognition of where the cases had occurred, as the mesh block IDs need a dedicated index to relate back to the geography. In this way, the mesh block data of where BuruIi ulcer cases occurred was only obvious once the results of the script were evaluated in the form of AUC metrics and subsequently plotted using QGIS software.

4. Comparing the summer and winter surveys at the Mornington Peninsula, the distribution of M. ulcerans positive excreta appears to have changed quite substantially, especially given that the possums are reported to be highly territorial with a range of only 100m. This version of the manuscript does not formally compare these spatial distributions, only the averages. Such an analysis would help understand if it is the possums that are moving, whether the possums undergo 'waves' of carriage (or indeed any other explanation), or if these apparent differences are down to chance.

Indeed, the spatial distribution IS*2404* positive possum excreta shifted to some extent between the southern hemisphere’s summer and winter suggesting possum populations might undergo waves of carriage over time. Taking the reviewer’s suggestion, we have explored this observation further in an additional supplemental figure (line 510, Figure S7). While this is an interesting observation, an in-depth assessment of seasonal carriage differences is beyond the scope and objectives of this current study.

5. Line 75. "Gut carriage" of Mu by possums is assumed but has never been strictly proven. Successful culture of M. ulcerans from a possum or its excreta has yet to be achieved. While this is likely a technical limitation due to the bacteria's extremely slow doubling time, it should be transparently discussed for the benefit of the wider readership audience of eLife.

We have moderated this sentence as suggested (line 89 onwards). However, there is growing evidence to support a gut carriage hypothesis. We refer to the 2013 study from O’Brien and colleagues (manuscript ref. 6) who trapped and examined 27 possums. They reported culture of *M. ulcerans* from the liver, spleen and caudal abdominal cavity, and strong IS*2404* PCR positive from possum gut contents. A recent environmental survey used 16SrRNA PCR to assess bacterial viability and noted 65% of IS*2404* positive possum faecal samples were also positive by this viability assay (manuscript ref. 13).

6. Line 216. More information about how the clinical information was geocoded is required, especially where cases were non-resident. Does the clinical information include information on dates and locations of visits to the area? This isn't clear. Please also provide additional information about how the researchers were blinded to the clinical and surveillance data and at what point they were unblinded to each other's findings.

These details were provided in the methods section (line 267 onwards):

“Patients were suspected of having been infected with *M. ulcerans* in the Mornington Peninsula if they were either a resident of the peninsula or if they visited the area and had not reported recent (<12 months) contact with any other known BU endemic areas in the state”.

Regarding blinding, refer to response to query 4 above, where the mesh block data (i.e. patient location) were loaded into an R script as numeric ID string and thus there was no researcher recognition of where the cases had occurred, as the mesh block IDs need a dedicated index to relate back to the geography.

7. Line 221: Inferring from Figures S4 and S6, it appears that the exposure window is defined as the quoted IQR from ref [4] of 101-171 days. If this is correct, please state it clearly, and expand more on how these values were arrived at. Is it possible to test the effect on the statistical model if this window is adjusted? A window that increases the model's predictive power might be informative with regard to the (still open) question of the BU incubation period.

Please see our reply to query 3 above. We have re-run the model with the absolute range of observed exposure periods and noted a decrease in model performance with an extended incubation period.

8. Line 226. Please confirm that for all patients, it is recorded/known that patients had visited the mesh box assigned to them in the geocode within the exposure period defined in Figure S4, and clearly explain how any instances of patients vising more than one mesh box (if this occurred) was handled in the model.

The Health Protection Branch of the Victorian State Government Department of Health conducts telephone interviews with confirmed BU patients to investigate likely origin of acquisition as described previously (DOI: 10.33321/cdi.2020.44.93, DOI: 10.3201/eid2411.171593). This information is recorded in a central database. Using this data source, patients who were non-residents and with exposure to multiple Buruli ulcer endemic areas were excluded from this current study. However, there always remain some underlying uncertainties, as patient recollections of travel history can be varied, and we can usually never know exactly where someone acquired their infection, as we mention in the discussion (line 705 onwards).

Editorial/presentation of data:eLife recommends a particular colour palette to be fully accessible (Colour Universal Design https://jfly.uni-koeln.de/color/). Several of the figures are quite hard to see even for a non-colourblind person, given the amount of information packed into them. Giving some thought to the best colour scheme would enhance the overall readability of the manuscript.Including some subheadings in the results would also enhance the overall readability.

We have re-drafted figures as detailed below.

Figure S1. Please indicate Belmont in panel E.

The mesh block outlines of the suburb of Belmont are now rendered in blue in Panel E of Figure S1.

Figure S3. This legend is inadequate. It should be possible to understand what the figure shows without referring to the text. Presumably, it is the incidence of BU, but more details are needed, and this should be clearly described.

We have now expanded the description of this figure in the legend.

Line 354. Comment on how the positivity rate for Mu compares to other prior possum excreta surveys in the area.

The only other excreta survey conducted in this region returned a *M. ulcerans* positivity rate of 9.3% (manuscript ref. 12). We have included this information now in the discussion (line 677).

Figure 1. Rather a lot of samples have a Ct above 37. Have the authors tested in the model how changes in the PCR cut-off impact the prediction ability? This legend is also inadequate, as it appears to be solely a title. Ct value is not DNA concentration. The statistical test is not mentioned, please include this and confirm that it can apply to log2 data. Genocopies per sample might be a better metric here.

Refer to our response to query 1 regarding the IS*2404* qPCR Ct threshold. As we outline above, we have high confidence in the diagnostic yield of the IS*2404* qPCR, even at high Cts. We have not changed the units to genocopies, as this is a derivative of Ct and would add no additional value to this comparative analysis and this figure. We have expanded the legend with more detail to help interpret the figure (line 464).

Figures 2 and 3. Very hard to visualise the mesh blocks in this figure due to the similar colour of the background to the BU case scale (see my general comment above).

The fill and outline of mesh blocks without cases was updated in Figures 2, 3 and S8. The selected colour scheme will now not confuse people with red-green colour blindness according to colour brewer (https://colorbrewer2.org).

Figure S4 and S6 would sit better with the methodology, as it is highly relevant to the section lines 221-228.

These supplemental figures have now been moved up to the Methods section.

Figure 5. Blue/red horribly unclear at the resolution provided. A better colour choice is needed.

We have re-drawn this figure in greyscale to improve legibility.

Line 519. This would be a good place to briefly acknowledge the limitations of the PCR approach used.

We have added a small section to the discussion to address this issue (line 741).

Line 530. Good point to comment on differences in the spatial distribution in the summer and winter surveys.

We have added section to the discussion to suggest that possum excreta surveys should be conducted over a time-period where transmission risk is known to be highest as the pattern of *M. ulcerans* excreta positivity can change between seasons (line 676 onwards).

Discussion point: Since the mosquito seems to be an important mechanical vector for transporting the Mu from possum excreta into humans, their role in this seems currently under-discussed. In particular, what is the range of a mosquito? Is this known? Could this influence the findings?

As we acknowledge in the discussion, mosquitoes likely play an important role in the spread of Buruli ulcer in southeast Australia. However, mosquitoes are not the focus of this research and will be dealt with in subsequent publications. The flight range of mosquitoes in this region was recently published and can be kilometres [DOI: 10.1038/s41437-022-00584-4].

Discussion point: It is important to include a qualifying statement that the model developed is only applicable to Australia, since no link is currently known between animal excreta or mosquitos in other countries, particularly West Africa where the number of BU cases is highest.

A sentence has been added to the end of the discussion (line 780).

Replace BU transmission with M. ulcerans transmission in the manuscript. BU is the resulting disease.

These instances have been corrected.

Abstract /elsewhere: Is really Buruli ulcer a "neglected disease"? Is it really contributive to "One-Health" in that topic?

WHO recognises Buruli ulcer as a neglected disease. We argue that any example of an emerging human disease with a major wildlife reservoir will be contributive to a One Health framework.

Abstract, line 37: Clarify the temporality of Buruli ulcer cases/faeces investigation campaign.

We have added a phrase (line 27).

Line 63: Were they microbiologically documented cases?

Yes – all cases were laboratory confirmed. This point is detailed in the research cited.

Line 140-174: Any negative control faeces (from animal demonstrated as feces-negative in previous, referenced studies?).

We used sterile water as our extraction negative control material, included blinding and at a frequency of 5-10% in each batch of 96 samples processed for extraction and qPCR (refer Figure S3 and methods).

Line 158-160: Was morphological identification, previously validated? Or would genetic identification be an added value?

We have previously attempted genetic ID from possum excreta, but there is very little host DNA in this specimen, making the PCR unreliable. Morphological ID is quite reliable as the excreta from the two possum species have very distinct size and shape characteristics (Figure S2).

Line 188-189: Clarify the number of positive/negative controls per experimental batch. Clarify the M. ulcerans- specificity of IS2404.

DNA extraction negative controls were included randomly at a frequency of 5-10% per batch of 96 samples and processed blinded. Additionally, every real-time PCR run included a no-template control (NTC) and a positive *M. ulcerans* control. This is now clarified in this methods section. We also included an additional schematic 96-well plate stack of all experimental controls used during 30 separate extraction and qPCR runs as a supplemental figure (line 220 onwards, Figure S3).

Line 213: "… detected laboratories…": please correct.

Corrected

Line 345-357: Give results for negative controls. Give results for positive controls, clarifying the interest of such positive controls for this study.

We added a paragraph in the relevant section reporting the outcome of all used experimental controls. We now also discuss the implication of this to the molecular detection assay’s overall accuracy and reliability (line 425 onwards).

Line 526: The authors may want to expand this point relative to "reservoir" (although it was not in the spectrum of the study), rather than "source" of infection for the population.

We weren’t clear what was being requested here. A discussion of sources and sinks from an ecological perspective is perhaps better placed in a subsequent review of *M. ulcerans* transmission.

Authors may want to clarify the specific contribution of each author.

We have added an author contribution section (lines 792-795)